# Focus on the Core: Efficient Attention via Pruned Token Compression for Document Classification

**Jungmin Yun[1]**         **Mihyeon Kim[1]**         **Youngbin Kim[1, 2]**

[1]Department of Artificial Intelligence, Chung-Ang University
[2]Graduate School of Advanced Imaging Sciences, Multimedia and Film, Chung-Ang University
{cocoro357, mh10967, ybkim85}.cau.ac.kr

## Abstract

Transformer-based models have achieved dominant performance in numerous NLP tasks. Despite their remarkable successes, pre-trained transformers such as BERT suffer from a computationally expensive self-attention mechanism that interacts with all tokens, including the ones unfavorable to classification performance. To overcome these challenges, we propose integrating two strategies: token pruning and token combining. Token pruning eliminates less important tokens in the attention mechanism's key and value as they pass through the layers. Additionally, we adopt fuzzy logic to handle uncertainty and alleviate potential mispruning risks arising from an imbalanced distribution of each token's importance. Token combining, on the other hand, condenses input sequences into smaller sizes in order to further compress the model. By integrating these two approaches, we not only improve the model's performance but also reduce its computational demands. Experiments with various datasets demonstrate superior performance compared to baseline models, especially with the best improvement over the existing BERT model, achieving +5%$p$ in accuracy and +5.6%$p$ in F1 score. Additionally, memory cost is reduced to 0.61x, and a speedup of 1.64x is achieved.

## 1 Introduction

Transformer-based deep learning architectures have achieved dominant performance in numerous areas of natural language processing (NLP) studies (Devlin et al., 2018; Lewis et al., 2019; Brown et al., 2020; Yang et al., 2019). In particular, pre-trained transformer-based language models like BERT (Devlin et al., 2018) and its variants (Yasunaga et al., 2022; He et al., 2020; Guo et al., 2020) have demonstrated state-of-the-art performance on many NLP tasks. The self-attention mechanism, a key element in transformers, allows for interactions between every pair of tokens in a sequence. This effectively captures contextual information across the entire sequence. This mechanism has proven to be particularly beneficial for text classification tasks (Yang et al., 2020; Karl and Scherp, 2022; Munikar et al., 2019).

Despite their effectiveness, BERT and similar models still face major challenges. BERT can be destructive in that not all tokens contribute to the final classification prediction (Guan et al., 2022). Not all tokens are attentive in multi-head self-attention, and uninformative or semantically meaningless parts of the input may not have a positive impact on the prediction (Liang et al., 2022). Further, the self-attention mechanism, which involves interaction among all tokens, suffers from substantial computational costs. Its quadratic complexity relative to the length of the input sequences results in high time and memory costs, making training impractical, especially for document classifications (Lee et al., 2022; Pan et al., 2022). In response to these challenges, many recent studies have attempted to address the problem of computational inefficiency and improve model ability by focusing on a few core tokens, thereby reducing the number of tokens that need to be processed. Their intuition is similar to human reading comprehension achieved by paying closer attention to important and interesting words (Guan et al., 2022).

One approach is a pruning method that removes a redundant token. Studies have shown an acceptable trade-off between performance and cost by simply removing tokens from the entire sequence to reduce computational demands (Ma et al., 2022; Goyal et al., 2020; Kim and Cho, 2020). However, this method causes information loss, which degrades the performance of the model (Wei et al., 2023). Unlike previous studies, we apply pruning to remove tokens from the keys and values of the attention mechanism to prevent the information loss and reduce the cost. In our method, less important tokens are removed, and the number of tokens gradually decreases by a certain ratio as they pass

through the layers. However, there is still a risk of mispruning when the distribution of importance is imbalanced, since the ratio-based pruning does not take into account the importance distribution (Zhao et al., 2019). To address this issue, we propose to adopt the fuzzy logic by utilizing fuzzy membership functions to reflect the uncertainty and support token pruning.

However, the trade-off between performance and cost of pruning limits the number of tokens that can be removed, hence, self-attention operations may still require substantial time and memory resources. For further model compression, we propose a token combining approach. Another line of prior works (Pan et al., 2022; Chen et al., 2023; Bolya et al., 2022; Zeng et al., 2022) have demonstrated that combining tokens can reduce computational costs and improve performance in various computer vision tasks, including image classification, object detection, and segmentation. Motivated by these studies, we aim to compress text sequence tokens. Since text differs from images with locality, we explore Slot Attention (Locatello et al., 2020), which can bind any object in the input. Instead of discarding tokens from the input sequence, we combine input sequences into smaller number of tokens adapting the Slot Attention mechanism. By doing so, we can decrease the amount of memory and time required for training, while also minimizing the loss of information.

In this work, we propose to integrate token pruning and token combining to reduce the computational cost while improving document classification capabilities. During the token pruning stage, less significant tokens are gradually eliminated as they pass through the layers. We implement pruning to reduce the size of the key and value of attention. Subsequently, in the token combining stage, tokens are merged into a combined token. This process results in increased compression and enhanced computational efficiency.

We conduct experiments with document classification datasets in various domains, employing efficient transformer-based baseline models. Compared to the existing BERT model, the most significant improvements show an increase of $5\%p$ in accuracy and an improvement of $5.6\%p$ in the F1 score. Additionally, memory cost is reduced to 0.61x, and a speedup of 1.64x is achieved, thus accelerating the training speed. We demonstrate that our integration results in a synergistic effect

not only improving performance, but also reducing memory usage and time costs.

Our main contributions are as follows:

- We introduce a model that integrates token pruning and token combining to alleviate the expensive and destructive issues of self-attention-based models like BERT. Unlike previous works, our token pruning approach removes tokens from the attention's key and value, thereby reducing the information loss. Furthermore, we use fuzzy membership functions to support more stable pruning.

- To our knowledge, our token combining approach is the first attempt to apply Slot Attention, originally used for object localization, for lightweight purposes in NLP. Our novel application not only significantly reduces computational load but also improves classification performance.

- Our experiment demonstrates the efficiency of our proposed model, as it improves classification performance while reducing time and memory costs. Furthermore, we highlight the synergy between token pruning and combining. Integrating them enhances performance and reduces overall costs more effectively than using either method independently.

## 2 Related Works

### 2.1 Sparse Attention

In an effort to decrease the quadratic time and space complexity of attention mechanisms, sparse attention sparsifies the full attention operation with complexity $O(n^2)$, where $n$ is the sequence length. Numerous studies have addressed the issue of sparse attention, which can hinder the ability of transformers to effectively process long sequences. The studies also demonstrate strong performances, especially in document classification. Sparse Transformer (Child et al., 2019) introduces sparse factorizations of the attention matrix by using a dilated sliding window, which reduces the complexity to $O(n\sqrt{n})$. Reformer (Kitaev et al., 2020) reduces the complexity to $O(nlogn)$ using the locality-sensitive hashing attention to compute the nearest neighbors. Longformer (Beltagy et al., 2020) scales complexity to $O(n)$ by combining local window attention with task-motivated global attention,

making it easy to process long documents. Linformer (Wang et al., 2020) performs linear self-attention with a complexity of $O(n)$, theoretically and empirically showing that self-attention can be approximated by a low-rank matrix. Similar to our work, Linformer reduces the dimensions of the key and value of attention. Additionally, we improve the mechanism by reducing the number of tokens instead of employing the linear projection, to maintain the interpretability. BigBird (Zaheer et al., 2020) introduces a sparse attention method with $O(n)$ complexity by combining random attention, window attention, and global attention. BigBird shows good performance on various long-document tasks, but it also demonstrates that sparse attention mechanisms cannot universally replace dense attention mechanisms, and that the implementation of sparse attention is challenging. Additionally, applying sparse attention has the potential risks of incurring context fragmentation and leading to inferior modeling capabilities compared to models of similar sizes (Ding et al., 2020).

## 2.2 Token Pruning and Combining

Numerous studies have explored token pruning methods that eliminate less informative and redundant tokens, resulting in significant computational reductions in both NLP (Kim et al., 2022; Kim and Cho, 2020; Wang et al., 2021) and Vision tasks (Chen et al., 2022; Kong et al., 2021; Fayyaz et al., 2021; Meng et al., 2022). Attention is one of the active methods used to determine the importance of tokens. For example, PPT (Ma et al., 2022) uses attention maps to identify human body tokens and remove background tokens, thereby speeding up the entire network without compromising the accuracy of pose estimation. The model that uses the most similar method to our work to determine the importance of tokens is LTP (Kim et al., 2022). LTP applies token pruning to input sequences in order to remove less significant tokens. The importance of each token is calculated through the attention score. On the other hand, DynamicViT (Rao et al., 2021) proposes an learned token selector module to estimate the importance score of each token and to prune less informative tokens. Transkimmer (Guan et al., 2022) leverages the skim predictor module to dynamically prune the sequence of token hidden state embeddings. Our work can also be interpreted as a form of sparse attention that reduces the computational load of

attention by pruning the tokens. However, there is a limitation to pruning mechanisms in that the removal of tokens can result in a substantial loss of information (Kong et al., 2021).

To address this challenge, several studies have explored methods for replacing token pruning. ToMe (Bolya et al., 2022) gradually combines tokens based on their similarity instead of removing redundant ones. TokenLearner (Ryoo et al., 2021) extracts important tokens from visual data and combines them using MLP to decrease the number of tokens. F-TFM (Dai et al., 2020) gradually compresses the sequence of hidden states while still preserving the ability to generate token-level representations. Slot Attention (Locatello et al., 2020) learns a set of task-dependent abstract representations, called "slots", to bind the objects in the input through self-supervision. Similar to Slot Attention, GroupViT (Xu et al., 2022) groups tokens that belong to similar semantic regions using cross-attention for semantic segmentation with weak text supervision. In contrast to GroupViT, Slot Attention extracts object-centric representations from perceptual input. Our work is fundamentally inspired by Slot Attention. To apply Slot Attention, which uses a CNN as the backbone, to our transformer-based model, we propose a combining module that functions similarly to the grouping block of GroupViT. TPS (Wei et al., 2023) introduces an aggressive token pruning method that divides tokens into reserved and pruned sets through token pruning. Then, instead of removing the pruned set, it is squeezed to reduce its size. TPS shares similarities with our work in that both pruning and squeezing are applied. However, while TPS integrates the squeezing process with pruning by extracting information from pruned tokens, our model processes combining and pruning independently.

## 3 Methods

In this section, we first describe the overall architecture of our proposed model, which integrates token pruning and token combining. Then, we introduce each compression stage in detail, including the token pruning strategy in section 3.2 and the token combining module in section 3.3.

## 3.1 Overall Architecture

Our proposed model architecture is illustrated in Figure 1. The existing BERT model (Devlin et al.,

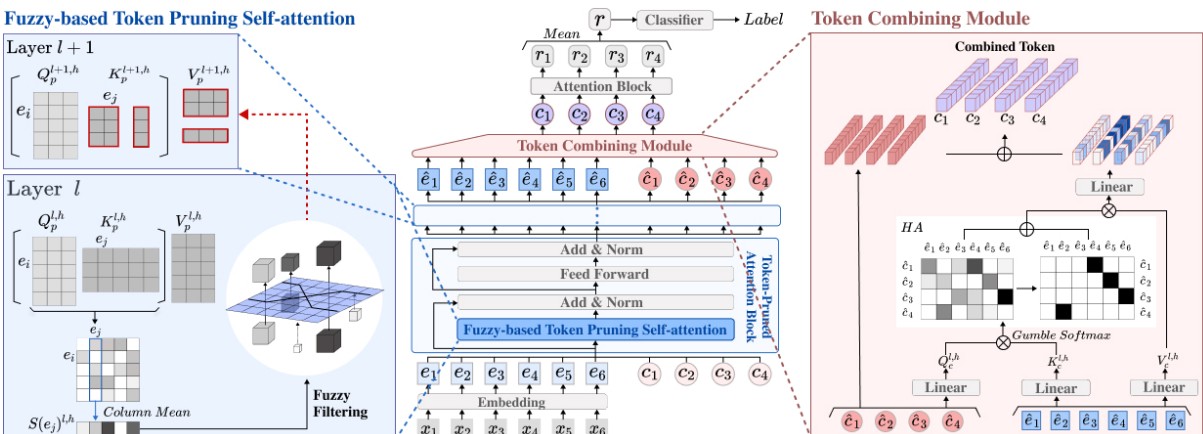

Figure 1: **Overall architecture of our purposed model** Model architecture is composed of several Token-pruned Attention Blocks, a Token Combining Module, and Attention Blocks. (Left): **Fuzzy-based Token Pruning Self-attention** In each layer, fuzzy-based pruning method removes tokens using importance score and fuzzy membership function. (Right): **Token Combining Module** This module apportions embedded tokens to each of the combination token using a similarity matrix between them.

2018) consists of stacked attention blocks. We modify the vanilla self-attention mechanism by applying a fuzzy-based token pruning strategy. Subsequently, we replace one of the token-pruned attention blocks with a token combining module. Replacing a token-pruned attention block instead of inserting an additional module not only enhances model performance but also reduces computational overhead due to its dot product operations. First, suppose $X = \{x_i\}_{i=1}^n$ is a sequence token from an input text with sequence length $n$. Given $X$, let $E = \{e_i\}_{i=1}^n$ be an embedded token after passing through the embedding layer. Each $e_i$ is an embedded token that corresponds to the sequence token $x_i$. Additionally, we add learnable combination tokens. Suppose $C = \{c_i\}_{i=1}^m$ is a set of learnable combination tokens, where $m$ is the number of combination tokens. These combination tokens bind other embedded tokens through the token combining module. We simplify $\{e_i\}_{i=1}^n$ to $\{e_i\}$ and $\{c_i\}_{i=1}^m$ to $\{c_i\}$. We concatenate $\{e_i\}$ and $\{c_i\}$ and use them as input for token-pruned attention blocks. We denote fuzzy-based token pruning self-attention by $FTP_{Attn}$ , feed-forward layers by $FF$, and layer norm by $LN$. The operations performed within the token-pruned attention block in $l$-th layer are as follows:

$$\{\tilde{e}_i^l\}, \{\tilde{c}_i^l\} = FTP_{Attn}([\{e_i^l\}; \{c_i\}^l]) \quad (1)$$

$$\{\hat{e}_i^l\}, \{\hat{c}_i^l\} = LN(FF([\{\tilde{e}_i^l\}; \{\tilde{c}_i^l\}]) + [\{\tilde{e}_i^l\}; \{\tilde{c}_i^l\}] \quad (2)$$

The token combining module receives $\{\hat{e}_i^l\}$ and $\{\hat{c}_i^l\}$ as input and merges $\{\hat{e}_i^l\}$ into $\{\hat{c}_i^l\}$ to output combined tokens $\{c_i^{l+1}\}$. After the token combining module, subsequent attention blocks do not perform pruning. Finally, we obtain the sequence representation by aggregating the output tokens $\{r_i\}$, in which our method averages the output.

### 3.2 Fuzzy-based Token Pruning Self-attention

We modify vanilla self-attention by implementing token pruning. Our token pruning attention mechanism gradually reduces the size of the key and value matrices by eliminating relatively unimportant embedded tokens, except for the combination tokens.

**Importance Score** We measure the significance of tokens based on their importance score. For each layer and head, the attention probability $Attention_{prob}$ is defined as:

$$Attention_{prob}^{l,h} = softmax(\frac{Q_p^{l,h} K_p^{l,h^T}}{\sqrt{d}}) \in \mathbb{R}^{n \times n} \quad (3)$$

where $l$ is the layer index, $h$ is the head index, $d$ is the feature dimension, and $Q_p^{l,h}, K_p^{l,h} \in \mathbb{R}^{n \times \frac{d}{h}}$ indicate query, key, and respectively. $Attention_{prob}$ is interpreted as a similarity between the $i$-th token $e_i$ and the $j$-th token $e_j$ , with row index $i \in [1, n]$ and column index $j \in [1, n]$. As the similarity increases, a larger weight is assigned to the value corresponding to $e_j$ . The $j$-th column in Equation

3 represents the amount of token $e_j$ attended by other tokens $e_i$ (Wang et al., 2021). Therefore, $e_j$ is considered a relatively important token as it is attended by more tokens. We define the importance score $S(e_j)$ in layer $l$ and head $h$ as:

$$S(e_j)^{l,h} = \frac{1}{n}\sum_{i=1}^{n}(Attention_{prob}^{l,h})_{i,j} \quad (4)$$

**Token Preservation Ratio** After calculating the importance score using $Q_p$ and $K_p$ in the $l$-th layer, we select $t_{l+1}$ embedded tokens in descending order of their scores. The $t_{l+1}$ embedded tokens are then indexed for $K_p$ and $V_p$ in the $(l+1)$-th layer. Other embedded tokens with relatively low importance score are pruned as a result. We define the number of tokens that remain after token pruning in the $(l+1)$-th layer as:

$$t^{l+1} = \lfloor t^l \times p \rfloor \quad (5)$$

where $t_{l+1}$ depends on $p$, a hyperparameter indicating the token preservation ratio of $t^{l+1}$ to $t^l$. This preservation ratio represents the proportion of tokens that are retained after pruning, relative to the number of tokens before pruning. As token pruning is not performed in the first layer, $t_1 = n$, and the attention uses the entire token in $Q_p$, $K_p$, and $V_p$. In the $(l+1)$-th layer, tokens are pruned based on $S(e_j)^{l,h}$ with $Q_p^{l,h} \in \mathbb{R}^{n \times \frac{d}{h}}$ and $K_p^{l,h} \in \mathbb{R}^{t^l \times \frac{d}{h}}$, where $t^{l+1} \leq t^l$. In the subsequent layers, the dimensions of $K_p$ and $V_p$ gradually decreases.

**Fuzzy-based Token Pruning** However, simply discarding a fixed proportion of tokens based on a importance score could lead to mispruning. Especially in imbalanced distributions, this pruning strategy may remove crucial tokens while retaining unimportant ones, thereby decreasing the model accuracy (Zhao et al., 2019). Insufficient training in the initial layers of the model can lead to uncertain importance scores, thereby increasing the risk of mistakenly pruning essential tokens. Furthermore, the importance score of a token is relative, and the distinction between the degree of importance and unimportance may be unclear and uncertain. To address this challenge, we exploit fuzzy theory, which can better perceive uncertainty. We employ two fuzzy membership functions to evaluate the degree of importance and unimportance together. Inspired by the

previous work (Zhao et al., 2019) on fuzzy-based filter pruning in CNN, we design fuzzy membership functions for $Importance(S(e))$ and $Unimportance(S(e))$ as:

$$Importance(S) = \begin{cases} 0 & \text{if } S(e) \leq a \\ \frac{S(e)-a}{b-a} & \text{if } a < S(e) < b \\ 1 & \text{if } S(e) \geq b \end{cases} \quad (6)$$

$$Unimportance(S) = \begin{cases} 1 & \text{if } S(e) \leq a \\ \frac{b-S(e)}{b-a} & \text{if } a < S(e) < b \\ 0 & \text{if } S(e) \geq b \end{cases} \quad (7)$$

where we simplify the importance score $S(e_j)^{l,h}$ to $S(e)$. Unlike the previous work(Zhao et al., 2019) that uses fixed constants as hyperparameters, our approach adopts the quantile function $Q_{S(e)}(0.25)$ and $Q_{S(e)}(0.75)$ for $a$ and $b$, respectively, to ensure robustness. We compute a quantile function for all importance scores, capturing the complete spectrum of head information. The importance set $I$ and the unimportance set $U$ are defined using the $\alpha - cut$, commonly referred to as $^{\alpha}A = x|A(x) \geq \alpha$ in fuzzy theory. To mitigate information loss due to imbalanced distribution, we employ token pruning based on the preservation ratio $p$ for tokens that fall within the set $(I - U)^c$. In the initial layers, where attention might not be adequately trained, there's a risk of erroneously removing crucial tokens. To counteract this, we've set the $\alpha$ for $I$ to a minimal value of 0.01, while the $\alpha$ for $U$ is empirically set to 0.9. Finally, our fuzzy-based token pruning self-attention $FTP_{Attn}$ is defined as :

$$FTP_{Attn} = softmax(\frac{Q_p^{l,h}K_p^{l,h^T}}{\sqrt{d}})V_p^{l,h},$$
$$(Q_p^{l,h} \in \mathbb{R}^{n \times \frac{d}{h}}, \ K_p^{l,h}, V_p^{l,h} \in \mathbb{R}^{t^l \times \frac{d}{h}}) \quad (8)$$

### 3.3 Token Combining Module

Token combining module takes token-pruned attention block's output representation $\hat{e}_i^l$, $\hat{c}_i^l$ as inputs. Combination tokens, which are concatenated with embedded tokens, pass through token-pruned attention blocks to incorporate global information from input sequences. Then, combination tokens integrate embedded tokens based on their similarity in the embedded space. Similar to GroupViT (Xu et al., 2022), our token combining module uses

Gumbel-Softmax (Jang et al., 2016) to perform cross-attention between combination tokens and embedded tokens. We define the similarity matrix $Sim$ as:

$$Sim_{i,j}^l$$
$$= \frac{\exp(W_q LN(\hat{c}_i^l) \cdot W_k LN(\hat{e}_j^l)) + g_i}{\sum_{t=1}^m \exp(W_q LN(\hat{c}_t^l) \cdot W_k LN(\hat{e}_j^l)) + g_t} \quad (9)$$

where $LN$ is layer normalization, $W_q$ and $W_k$ are the weights of projection matrix for the combination tokens and embedded tokens, respectively, and $\{g_i\}$ are $i.i.d$ random samples from the $Gumbel(0,1)$ distribution. Subsequently, we implement hard assignment technique (Xu et al., 2022), which employs a one-hot operation to determine the specific combination token to which each embedded token belongs. We define hard assignment $HA$ as:

$$HA_{i,j}^l = \mathbb{1}_{M_i^l}(Sim_{i,j}^l) + Sim_{i,j}^l - sg(Sim_{i,j}^l),$$
$$M_i = max(Sim_{i,-}) \quad (10)$$

where $sg$ is the stop gradient operator to stop the accumulated gradient of the inputs. We update the combination token by calculating the weighted sum of the embedded token that corresponds to the same combination token. The output of the token combining block is calculated as follows:

$$c_i^{l+1} = \hat{c}_i^l + W_o \frac{\sum_{j=1}^m HA_{i,j}^l W_v \hat{e}_j^l}{\sum_{j=1}^m HA_{i,j}^l} \quad (11)$$

where $W_v$ and $W_o$ are the weights of the projection matrix. We adopt the grouping mechanism described in GroupViT. GroupViT learns semantic segmentation by grouping output segment tokens to object classes through several grouping stages. Our method, on the other hand, replaces one layer with a token combining module and compresses embedded tokens to a few informative combined tokens. Empirically, we find that this approach reduces the training memory and time of the model, increasing performance.

## 4 Experiments

This section aims to validate the effectiveness of our proposed model. Firstly, we evaluate the document classification performance of our proposed model compared to the baseline models. Secondly, we investigate the time and memory costs of our proposed model and evaluate its efficiency. Lastly, through the ablation study, we compare the effects of different preservation ratios on fuzzy-based token pruning self-attention. We also analyze the impact of the position of the token combining module and the number of combination tokens.

### 4.1 Dataset

We evaluate our proposed model using six datasets across different domains with varying numbers of classes. SST-2 (Socher et al., 2013) and IMDB (Maas et al., 2011) are datasets for sentiment classification on movie reviews. BBC News (Greene and Cunningham, 2006) and 20 NewsGroups (Lang, 1995) comprise a collection of public news articles on various topics. LEDGAR (Tuggener et al., 2020) includes a corpus of legal provisions in contract, which is part of the LexGLUE (Chalkidis et al., 2021) benchmark to evaluate the capabilities of legal text. arXiv is a digital archive that stores scholarly articles from a wide range of fields, such as mathematics, computer science, and physics. We use the arXiv dataset employed by the controller (He et al., 2019) and perform classification based on the abstract of the paper as the input. We present more detailed statistics of the dataset in Table 1.

| Dataset | Genre | $C$ | $I$ | $T$ |
|---|---|---|---|---|
| SST-2 | review | 2 | 9613 | 23.2 |
| IMDB | review | 2 | 50000 | 292.2 |
| BBC News | news | 5 | 2225 | 452.7 |
| 20 NewsGroup | news | 20 | 18846 | 330.6 |
| LEDGAR | legal | 100 | 80000 | 138.7 |
| arXiv | scientific publication | 11 | 1200 | 202.6 |

Table 1: Statistics of the datasets. $C$ denotes the number of classes in the dataset, $I$ the number of instances, and $T$ the average number of tokens calculated using BERT(`bert-base-uncased`) tokenizer.

### 4.2 Experimental Setup and Baselines

The primary aim of this work is to address the issues of the BERT, which can be expensive and destructive. To evaluate the effectiveness of our proposed model, we conduct experiments comparing it with the existing BERT model(`bert-base-uncased`). For a fair comparion, both of BERT and our proposed model follow the same settings, and ours is warm-started on `bert-base-uncased`. Our model has the same number of layers and heads, embedding and hid-

Table 2 (document classification performance comparison):

| Dataset | Model | Location | Accuracy | F1(macro) | F1(micro) | Dataset | Model | Location | Accuracy | F1(macro) | F1(micro) |
|---|---|---|---|---|---|---|---|---|---|---|---|
| SST-2 | BigBird | - | 91.8 | 91.8 | 91.8 | 20 NewsGroup | BigBird | - | 69.3 | 67.7 | 69.3 |
| | Longformer | - | 91.0 | 91.0 | 91.0 | | Longformer | - | 68.5 | 67.0 | 68.5 |
| | F-TFM | - | 92.0 | 92.0 | 92.0 | | F-TFM | - | 69.7 | 68.5 | 69.7 |
| | Transkimmer | - | 91.4 | 91.4 | 91.4 | | Transkimmer | - | 67.8 | 64.9 | 67.8 |
| | BERT | - | 90.7 | 89.4 | 90.7 | | BERT | - | 68.7 | 56.1 | 68.7 |
| | ours-P | - | 91.1 | 89.8 | 91.1 | | ours-P | - | 69.4 | 56.8 | 70.0 |
| | ours-PF | - | 91.4 | 90.1 | 91.4 | | ours-PF | - | **70.7** | **58.2** | **70.7** |
| | ours-C | layer 11 | 91.8 | 90.8 | 91.8 | | ours-C | layer 11 | 69.2 | 56.5 | 69.2 |
| | ours-PFC | layer 11 | **92.1** | **91.0** | **92.1** | | ours-PFC | layer 11 | 69.9 | 57.1 | 69.9 |
| | ours-PFC | layer 7 | 90.1 | 88.6 | 90.1 | | ours-PFC | layer 7 | 68.5 | 55.4 | 68.5 |
| IMDB | BigBird | - | 92.8 | 92.8 | 92.8 | LEDGAR | BigBird | - | 86.9 | 78.8 | 86.9 |
| | Longformer | - | 93.4 | 93.4 | 93.4 | | Longformer | - | 84.0 | 82.1 | 84.0 |
| | F-TFM | - | 91.7 | 91.7 | 91.7 | | F-TFM | - | 86.7 | 78.4 | 86.7 |
| | Transkimmer | - | 93.7 | 93.7 | 93.7 | | Transkimmer | - | 87.1 | 77.3 | 87.1 |
| | BERT | - | 93.7 | 92.8 | 93.7 | | BERT | - | 86.2 | 77.4 | 86.2 |
| | ours-P | - | 93.8 | 92.9 | 93.8 | | ours-P | - | 86.5 | 77.9 | 86.5 |
| | ours-PF | - | **94.3** | **93.4** | **94.3** | | ours-PF | - | 86.8 | 78.4 | 86.8 |
| | ours-C | layer 11 | 92.6 | 91.3 | 92.6 | | ours-C | layer 11 | 86.9 | 78.4 | 86.9 |
| | ours-PFC | layer 11 | 93.5 | 92.5 | 93.5 | | ours-PFC | layer 11 | **87.3** | **79.2** | **87.3** |
| | ours-PFC | layer 7 | 92.8 | 91.5 | 92.8 | | ours-PFC | layer 7 | 85.7 | 76.8 | 85.7 |
| BBC News | BigBird | - | 97.1 | 97.1 | 97.1 | arXiv | BigBird | - | 74.0 | 70.4 | 74.0 |
| | Longformer | - | 97.9 | 97.9 | 97.9 | | Longformer | - | 66.0 | 64.8 | 66.0 |
| | F-TFM | - | 96.5 | 96.5 | 96.5 | | F-TFM | - | 70.0 | 66.5 | 70.0 |
| | Transkimmer | - | 97.6 | 97.6 | 97.6 | | Transkimmer | - | 73.7 | 72.6 | 73.7 |
| | BERT | - | 96.2 | 94.0 | 96.2 | | BERT | - | 69.0 | 52.5 | 68.3 |
| | ours-P | - | 97.2 | 95.4 | 97.2 | | ours-P | - | 71.0 | 56.6 | 70.2 |
| | ours-PF | - | 97.9 | 96.6 | 97.9 | | ours-PF | - | 76.0 | **61.0** | **74.0** |
| | ours-C | layer 11 | 96.8 | 95.2 | 96.8 | | ours-C | layer 11 | 70.0 | 52.7 | 69.2 |
| | ours-PFC | layer 11 | **98.1** | **97.1** | **98.1** | | ours-PFC | layer 11 | 74.0 | 58.1 | 73.1 |
| | ours-PFC | layer 7 | 97.0 | 95.2 | 97.0 | | ours-PFC | layer 7 | 68.0 | 50.7 | 67.3 |

Table 2: Performance comparison on document classification. To our proposed model, *ours-P* applies token pruning, *ours-PF* applies fuzzy-based token pruning, *ours-C* applies token combining module, and *ours-PFC* applies both fuzzy-based token pruning and token combining module. The best performance is highlighted in **bold**.

| Model | Location | FLOPs | Memory Cost | Speedup |
|---|---|---|---|---|
| BigBird | - | 1.57x | 0.82x | 0.94x |
| Longformer | - | 2.10x | 1.11x | 0.55x |
| F-TFM | - | 1.03x | 0.39x | 1.11x |
| Transkimmer | - | 0.13x | 0.87x | 0.70x |
| BERT | - | - | - | - |
| ours-P | - | 1x | 0.88x | 1.33x |
| ours-PF | - | 1x | 0.88x | 1.25x |
| ours-C | layer 11 | 0.877x | 0.89x | 1.26x |
| ours-PFC | layer 11 | 0.877x | 0.80x | 1.29x |
| ours-PFC | layer 7 | 0.544x | 0.61x | 1.64x |

Table 3: Efficiency comparison on document classification.

den sizes, and dropout ratio as BERT. Our model has the same number of layers and heads, embedding and hidden sizes, and dropout ratio as BERT. We also compare our method to other baselines, including BigBird (Zaheer et al., 2020) and Longformer (Beltagy et al., 2020), which employ sparse attention, as well as F-TFM (Dai et al., 2020) and Transkimmer (Guan et al., 2022), which utilize token compression and token pruning, respectively.

For IMDB, BBC News, and 20NewsGroup, 20% of the training data is randomly selected for validation. During training, all model parameters are fine-tuned using the Adam optimizer (Kingma and

Ba, 2014). The first 512 tokens of the input sequences are processed. The learning rate is set to 2e-5, and we only use 3e-5 for the LEDGAR dataset. We also use a linear warm-up learning rate scheduler. In all experiments, we use a batch size of 16. We choose the model with the lowest validation loss during the training step as the best model. We set the token preservation ratio $p$ to 0.9.

## 4.3 Main Result

To evaluate the performance and the efficiency of each strategy, we compare our proposed model(*ours-PFC*) with five baselines and three other models: one that uses only token pruning(*ours-P*), one that applies fuzzy membership functions for token pruning(*ours-PF*), and one that uses only a token combining module(*ours-C*), as shown in Table 2 and Table 3.

Compared to BERT, *ours-P* consistently outperforms it for all datasets with higher accuracy and F1 scores, achieving approximately 1.33x speedup and 0.88x memory savings. More importantly, the performance of *ours-PF* significantly surpasses that of *ours-P* with up to 5.0%$p$ higher accuracy, 4.4%$p$ higher F1(macro) score, and 3.8%$p$ higher

F1(micro) score, with the same FLOPs and comparable memory costs. To evaluate the performance of *ours-C* and *ours-PFC*, we incorporate the combining module at the 11-th layer, which results in the optimal performance. A comprehensive discussion on performance fluctuations in relation to the location of the combining module is presented in Section 4.4. Excluding the IMDB dataset, *ours-C* not only achieves higher values in both the accuracy and F1 scores compared to the BERT but also exceeds by 0.89x speedup and 1.26x memory savings while reducing FLOPs. Across all datasets, our models(*ours-PF* or *ours-PFC*) consistently outperform all efficient transformer-based baseline models. Furthermore, *ours-PFC* outperforms the BERT with up to 5.0%$p$ higher accuracy, 5.6%$p$ higher macro F1 score, and 4.8%$p$ higher micro F1 score. Additionally, *ours-PFC* exhibits the best performance with the least amount of time and memory required, compared to models that use pruning or combining methodologies individually. These findings highlight the effectiveness of integrating token pruning and token combining on BERT's document classification performance, from both the performance and the efficiency perspective.

Subsequently, we evaluate the potential effectiveness of *ours-C* and *ours-PFC* by implementing the combining module at the 7-th layer. As shown in Table 5 of Section 4.4, applying the combining module to the 7-th layer leads to further time and memory savings while also mitigating the potential decrease in accuracy. Compared to BERT, it only shows a minimal decrease in accuracy (at most 0.8%$p$). Moreover, it reduces FLOPs and memory costs to 0.61x, while achieving a 1.64x speedup. In our experiments, we find that our proposed model effectively improves document classification performance, outperforming all baselines. Even when the combination module is applied to the 7th layer, it maintains performance similar to BERT while further reducing FLOPs, lowering memory usage, and enhancing speed.

## 4.4   Ablation study

**Token Preservation Ratio** We evaluate different token preservation ratios $p$ on the BBC News dataset, as shown in Table 4. Our findings indicate that the highest accuracy, at 98.1%, was achieved when $p$ is 0.9. Moreover, our fuzzy-based pruning strategy results in 1.17x reduction in memory cost and 1.12x speedup compared to the vanilla self-

attention mechanism. As $p$ decreases, we observe that performance deteriorates and time and memory costs decrease. A smaller $p$ leads to the removal of more tokens from the fuzzy-based token pruning self-attention. While leading a higher degree of information loss in attention, it removes more tokens can result in time and memory savings.

| Preservation Ratio $p$ | Accuracy | Memory Cost | Speedup |
|---|---|---|---|
| 0.95 | 97.7 | 0.99x | 1.07x |
| 0.9 | **97.9** | 0.86x | 1.12x |
| 0.85 | 97.6 | 0.79x | 1.14x |
| 0.8 | 97.2 | 0.75x | 1.17x |
| 0.75 | 97.4 | 0.71x | 1.19x |
| 0.7 | 97.2 | 0.69x | 1.24x |
| 0.65 | 96.8 | 0.68x | 1.25x |
| 0.6 | 96.5 | 0.67x | 1.25x |
| BERT | 96.2 | - | - |

Table 4: Comparisons of different token preservation ratios

**Token Combining Module** In Table 5, we compare the positions of different token combining modules on the BBC News dataset. We observe that placing the token combining module earlier within the layers results in greater speedup and memory savings. Since fewer combined tokens proceed through subsequent layers, the earlier this process begins, the greater the computational reduction that can be achieved. However, this reduction in the interaction between the combination token and the embedded token hinders the combination token to learn global information, potentially degrading the model performance. Our proposed model shows the highest performance when the token combining module is placed in the 11-th layer, achieving an accuracy of 98.1%.

| Layer | Accuracy | Memory Cost | Speedup | FLOPs |
|---|---|---|---|---|
| 5 | 96.5 | 0.47x | 1.69x | 2.65x |
| 6 | 96.6 | 0.54x | 1.53x | 2.17x |
| 7 | 97.8 | 0.61x | 1.39x | 1.84x |
| 8 | 97.2 | 0.68x | 1.29x | 1.60x |
| 9 | 97.4 | 0.76x | 1.17x | 1.41x |
| 10 | 97.8 | 0.82x | 1.17x | 1.26x |
| 11 | **98.1** | 0.89x | 1.03x | 1.14x |
| 11(*) | (97.7) | (0.90x) | (0.88x) | (1.14x) |
| 12 | 97.6 | 0.96x | 0.97x | 1.04x |

Table 5: Comparisons according to combining module location. We replace one of the existing transformer layers with a token combining module. (*) represents a case where a combining module is additionally inserted without replacing.

**Combination Tokens** In Table 6, we compare the accuracy achieved with different numbers of combination tokens. We use four datasets to analyze the impact of the number of classes on the combination tokens. We hypothesized that an increased number of classes would require more combination tokens to encompass a greater range of information. However, we observe that the highest accuracy is achieved when using eight combination tokens across all four datasets. When more combination tokens are used, performance gradually degrades. These results indicate that when the number of combination tokens is fewer than the number of classes, each combination token can represent more information as a feature vector in a 768-dimensional embedding space, similar to findings in GroupViT. Through this experiment, we find that the optimal number of combination tokens is 8. We show that our proposed model performs well in multi-class classification without adding computation and memory costs.

| Dataset | $C$ | Number of Combination Tokens | | | | |
|---------|-----|------|------|------|------|------|
|         |     | 4    | 8    | 16   | 32   | 64   |
| SST-2       | 2   | 91.2 | 92   | 90.9 | 91.2 | 89.7 |
| BBC News    | 5   | 97.3 | 98.1 | 97.5 | 97.6 | 97.2 |
| 20 NewsGroup | 20  | 68.5 | 69.9 | 68.9 | 68.7 | 67.6 |
| LEDGAR      | 100 | 85.6 | 87.3 | 86.7 | 86   | 85.9 |

Table 6: Performance comparison of different numbers of combination tokens. $C$ denotes the number of classes in the dataset.

## 5   Conclusion

In this paper, we introduce an approach that integrates token pruning and token combining to improve document classification by addressing expensive and destructive problems of self-attention in the existing BERT model. Our approach consists of fuzzy-based token pruned attention and token combining module. Our pruning strategy gradually removes unimportant tokens from the key and value in attention. Moreover, we enhance the robustness of the model by incorporating fuzzy membership functions. For further compression, our token combining module reduces the time and memory costs of the model by merging the tokens in the input sequence into a smaller number of combination tokens. Experimental results show that our proposed model enhances the document classification performance by reducing computational requirements with focusing on more significant tokens.

Our findings also demonstrate a synergistic effect by integrating token pruning and token combining, commonly used in object detection and semantic segmentation. Ultimately, our research provides a novel way to use pre-trained transformer models more flexibly and effectively, boosting performance and efficiency in a myriad of applications that involve text data processing.

## Limitations

In this paper, our goal is to address the fundamental challenges of the BERT model, which include high cost and performance degradation, that hinder its application to document classification. We demonstrate the effectiveness of our proposed method, which integrates token pruning and token combining, by improving the existing BERT model. However, our model, which is based on BERT, has an inherent limitation in that it can only handle input sequences with a maximum length of 512. Therefore, it is not suitable for processing datasets that are longer than this limit. The problems arising from the quadratic computation of self-attention and the existence of redundant and uninformative tokens are not specific to BERT and are expected to intensify when processing longer input sequences. Thus, we will improve other transformer-based models that can handle long sequence datasets, such as LexGLUE, and are proficient in performing natural language inference tasks in our future work.

## Ethics Statement

Our research adheres to ethical standards of practice. The datasets used to fine-tune our model are publicly available and do not contain any sensitive or private information. The use of open-source data ensures that our research maintains transparency. Additionally, our proposed model is built upon a pre-trained model that has been publicly released. Our research goal aligns with an ethical commitment to conserve resources and promote accessibility. By developing a model that minimizes hardware resource requirements and time costs, we are making a valuable contribution towards a more accessible and inclusive AI landscape. We aim to make advanced AI techniques, including our proposed model, accessible and practical for researchers with diverse resource capacities, ultimately promoting equity in the field.

## Acknowledgment

This research was supported by Basic Science Research Program through the National Research Foundation of Korea(NRF) funded by the Ministry of Education(NRF-2022R1C1C1008534), and Institute for Information & communications Technology Planning & Evaluation (IITP) through the Korea government (MSIT) under Grant No. 2021-0-01341 (Artificial Intelligence Graduate School Program, Chung-Ang University).

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
