# OpenReview forum: "Focus on the Core: Efficient Attention via Pruned Token Compression for Document Classification"
_EMNLP/2023/Conference — EMNLP 2023 Findings_

### Official Review · Reviewer_fmo5 · 2023-07-31

**Soundness:** 3

**Excitement:**

3: Ambivalent: It has merits (e.g., it reports state-of-the-art results, the idea is nice), but there are key weaknesses (e.g., it describes incremental work), and it can significantly benefit from another round of revision. However, I won't object to accepting it if my co-reviewers champion it.

**Paper Topic And Main Contributions:**

The paper aims at address the high cost and performance degradation of BERT models. The authors integrating token pruning and token combining. Token pruning prunes less important tokens and token combing further condense the inputs. Extensive experiments shows the effectiveness of the proposed methods with memory cost savings and speedup.

**Questions For The Authors:**

- Could you explain why fuzzy-based method work for pruning and give more analysis for how it works?
- How to decide whether a token belongs to unimportant set? And how do you set the pruning ratio $p$? Does the choice of $a$ and $b$ in equ. 6 and 7 impact the performance?

**Reasons To Accept:**

- Efficient models are a important direction.
- The experimental results are both effective and convincing, as they demonstrate the proposed methods' ability to achieve memory savings and speedup without compromising performance.

**Reasons To Reject:**

- The contribution is a combination of current methods from computer vision and incremental.
- The illustration in fig. 1 is confusing, the symbol definition is different from what the authors use in the text.
-  The motivation for fuzzy-based token pruning is not clear. Even in imbalanced distributions, discarding tokens according to the importance score will not discard important tokens while retaining unimportant tokens. Given that tokens of lesser significance inherently possess a lower ranking compared to their more important counterparts, the pruning process inherently eliminates tokens of smaller importance. So why introducing uncertainty in pruning? More uncertainty are more likely to drop important tokens.
- The fuzzy-based approach appears to limit the ability to distinguish between tokens characterized by high and low values. The alignment of the membership function with the concept of uncertainty remains unclear.
- The experiment comparison is weak, the author only compare their method to the BERT-baseline. The author should compare their method to token pruning and token combination baselines.

**Reproducibility:**

3: Could reproduce the results with some difficulty. The settings of parameters are underspecified or subjectively determined; the training/evaluation data are not widely available.

**Reviewer Confidence:**

4: Quite sure. I tried to check the important points carefully. It's unlikely, though conceivable, that I missed something that should affect my ratings.

---

> ### Author Rebuttal · Authors · 2023-08-29
>
> Thank you for your comprehensive review and valuable feedback on our manuscript. We recognize the importance of addressing the concerns you raised, especially regarding the fuzzy-based pruning method and the experimental comparison. Here's our response to your comments:
>
> ### Revised Manuscript
> Your feedback highlighted the need for a clearer exposition of our fuzzy-based pruning method. We acknowledge that the constraints of the paper’s length led us to provide a concise summary of the fuzzy-based pruning method, which may have caused some confusion. We deeply regret this oversight and aim to rectify it by providing a more detailed explanation.
>
> In the revised manuscript, particularly in lines [377–381], we have expanded our description as follows:
>
> > Unlike the previous work[11] that uses fixed constants as hyperparameters, our approach adopts the quantile functions $Q_{S(t)}(0.25)$ and $Q_{S(t)}(0.75)$ for $a$ and $b$, respectively, to ensure robustness. We compute a quantile function for all importance scores, capturing the complete spectrum of head information. The importance set $I$ and the unimportance set $U$ are defined using the $\alpha-cut$, commonly referred to as $^{\alpha}A={x|A(x) \geq \alpha}$ in fuzzy theory. To mitigate information loss due to imbalanced distribution, we employ token pruning based on the pruning ratio $p$ for tokens that fall within the set $(I-U)^c$. In the initial layers, where attention might not be adequately trained, there's a risk of erroneously removing crucial tokens. To counteract this, we've set the $\alpha$ for $I$ to a minimal value of 0.01, while the $\alpha$ for $U$ is empirically set to 0.9.
>
> We hope this expanded explanation provides a clearer understanding of our methodology.
>
> ### Response to Weakness 1:
> > The contribution is a combination of current methods from computer vision and incremental.
>
> Thank you for pointing out the perceived incremental nature of our work. We understand your concerns and would like to clarify the unique contributions of our approach, especially in the context of NLP.
>
> **Contribution Clarification**
>
> While it's true that our token combining module draws inspiration from grouping methods in computer vision, the application and challenges in the NLP domain are distinct. Our primary motivation is to address the computational and performance issues associated with self-attention rather than merely introduce a novel combining methodology.
>
> 1. **Fuzzy Logic in Token Pruning**: We've introduced fuzzy logic to enhance the existing token pruning strategies, which traditionally relied on straightforward attention scores. This addition is pivotal in ensuring that the pruning process is more adaptive and less prone to errors, especially in complex linguistic contexts.
>
> 2. **Token Combining in NLP**: The token combining strategy we've employed is significant in its own right. While similar methodologies might be prevalent in computer vision tasks like object detection or segmentation, their application to text classification in NLP is novel. We've successfully demonstrated its effectiveness in this new domain.
>
> 3. **Synergy between Pruning and Combining**: One of the standout features of our approach is the synergistic relationship between token pruning and combining. When integrated, these techniques lead to a model that's not only lightweight but also exhibits enhanced classification performance. Our experiments further validate this, showing marked improvements even when only the token combining module is applied.
>
> In essence, our work transcends the boundaries of methods traditionally confined to computer vision, showcasing their potential in NLP. This not only underscores the versatility of these techniques but also paves the way for future research in this direction.
>
> ### Response to Weakness 2:
> > The illustration in fig. 1 is confusing, the symbol definition is different from what the authors use in the text.
>
> Thank you for your feedback. We apologize for any confusion that our mistake may have caused. In the descriptions of Figure 1, (left) should be a  fuzzy-based token pruning self-attention, and (right) should be a token combining module. We have updated our descriptions in Figure 1 in our manuscripts.
>
> ### Response to Weakness 3:
> > The motivation for fuzzy-based token pruning is not clear. Even in imbalanced distributions, discarding tokens according to the importance score will not discard important tokens while retaining unimportant tokens. Given that tokens of lesser significance inherently possess a lower ranking compared to their more important counterparts, the pruning process inherently eliminates tokens of smaller importance. So why introducing uncertainty in pruning? More uncertainty are more likely to drop important tokens.
>
> **Motivation for Fuzzy-Based Token Pruning**
> We introduced fuzzy-based token pruning to address specific challenges that arise from the direct pruning method based on the importance score $S(e)$. Here are our primary motivations:
>
> 1. **Distribution of Importance Scores**: Direct pruning based on importance scores doesn't always account for the distribution of these scores across the dataset. In datasets dominated by crucial tokens, there's a higher likelihood of inadvertently pruning significant tokens. Our fuzzy-based approach mitigates this risk by considering the distribution of importance scores.
>
> 2. **Insufficient Learning in Shallow Layers**: In the initial layers of the model, the importance score $S(e)$ may not be adequately trained. This can lead to uncertainty in the importance scores, increasing the risk of pruning crucial tokens. Our fuzzy-based method addresses this challenge by introducing a more nuanced pruning strategy.
>
> **Addressing Uncertainty**
> Our approach doesn't aim to introduce uncertainty but rather to manage and mitigate it. We recognize that tokens with higher importance scores should be prioritized for preservation. However, due to the reasons mentioned above, there can be inherent uncertainties in the pruning process. By employing two fuzzy membership functions, we define the importance and unimportance sets for all representations. These sets act as filters, helping us identify the tokens that are most likely to be unimportant. Through this method, we ensure that vital information is retained, and our experiments validate the effectiveness of our fuzzy-based token pruning strategy.
>
> ### Response to Weakness 4:
> > The fuzzy-based approach appears to limit the ability to distinguish between tokens characterized by high and low values. The alignment of the membership function with the concept of uncertainty remains unclear.
>
> We acknowledge that our fuzzy membership functions, as defined in Eq. 6 and Eq. 7, may give the impression of limiting the ability to distinguish between tokens with high and low importance scores. However, this design is intentional and aims to address specific challenges:
>
> 1. **Managing Uncertainty**: As elaborated in "Response to Weekness 3," our fuzzy-based approach is designed to manage the uncertainties that arise from imbalanced data and insufficiently trained importance scores in shallow layers.
>
> 2. **Reliable Extremes**: Our fuzzy membership functions assign a probability value of $0$ or $1$ to extreme importance scores. This is based on the assumption that extreme scores are more reliable indicators of a token's importance or unimportance.
>
> 3. **Selective Pruning**: After defining the importance and unimportance sets using $\alpha-cut$, we apply pruning preferentially to the set deemed least important. This ensures that crucial information is retained while less important tokens are pruned.
>
> By doing so, our fuzzy-based approach provides a nuanced way of handling the complexities and uncertainties involved in token pruning, thereby improving the overall performance of the model.
>
> ### Response to Weekness 5:
> >The experiment comparison is weak, the author only compare their method to the BERT-baseline. The author should compare their method to token pruning and token combination baselines.
>
> Thank you for emphasizing the need for a broader experimental comparison. We've taken steps to address this:
>
> **Baselines**:
> - **Token Pruning**: We've introduced models like BigBird[1] and Longformer[2] which utilize sparse attention. Additionally, Funnel-Transformer[3] and Transkimmer[4] serve as examples of token compression and token pruning, respectively.
>
> - **Token Combination**: Our approach to token combinations using virtual tokens is novel. After extensive research, we found our paper to be a pioneering effort in this domain, especially for document classification tasks.
>
> **Performance Comparison on Document Classification:**
> |                        | **SST-2**       | **IMDB**        | **BBC News**    | **20NewsGroups** | **arXiv**       | **LEDGAR**      |
> |------------------------|-----------------|-----------------|-----------------|------------------|-----------------|-----------------|
> | **BigBird**            | 91.8 / 91.8     | 92.8 / 92.8     | 97.1 / 97.1     | 69.3 / 67.7      | 74.0 / 70.4     | 86.9 / 78.8     |
> | **Longformer**         | 91.0 / 91.0     | 93.4 / 93.4     | 97.9 / 97.9     | 68.5 / 67.0      | 66.0 / 64.8     | 84.0 / 82.1     |
> | **Funnel-Transformer** | 92.0 / 92.0     | 91.7 / 91.7     | 96.5 / 96.5     | 69.7 / 68.5      | 70.0 / 66.5     | 86.7 / 78.4     |
> | **Transkimmer**        | 91.4 / 91.4     | 93.7 / 93.7     | 97.6 / 97.6     | 67.8 / 64.9      | 73.7 / 72.6     | 87.1 / 77.3     |
> | **BERT**      | 90.7 / 89.4     | 93.7 / 92.8     | 96.2 / 94.0     | 68.7 / 56.1      | 69.0 / 52.5     | 86.2 / 77.4     |
> | **ours-P**             | 91.1 / 89.8     | 93.8 / 92.9     | 97.2 / 95.4     | 69.4 / 56.8      | 71.0 / 56.6     | 86.5 / 77.9     |
> | **our-PF**             | 91.4 / 90.1     | **94.3 / 93.4** | 97.9 / 96.6     | **70.7 / 58.2**  | **76.0 / 61.0** | 86.8 / 78.4     |
> | **ours-C**             | 91.8 / 90.8     | 92.6 / 91.3     | 96.8 / 95.2     | 69.2 / 56.5      | 70.0 / 52.7     | 86.9 / 78.4     |
> | **ours-PFC (L11)**     | **92.1 / 91.0** | 93.5 / 92.5     | **98.1 / 97.1** | 69.9 / 57.1      | 74.0 / 58.1     | **87.3 / 79.2** |
> | **ours-PFC (L7)**      | 90.1 / 88.6     | 92.8 / 91.5     | 97.0 / 95.2     | 68.5 / 55.4      | 68.0 / 50.7     | 85.7 / 76.8     |
>
> **Efficiency Comparison on Document Classification:**
>
> | Model              | FLOPs(↓)   |Memory Cost(↓)|  SpeedUp(↑) |
> | --------           | -------- | -----     | --------  |
> | **BERT**      |    (1.0x)    | (1.0x)         |  (1.0x)        |
> | **ours-PFC (L11)**     | 0.88x    | 0.80x     | 1.29x     |
> | **ours-PFC (L7)**      | 0.54x    | 0.61x     | 1.64x     |
> | **BigBird**            | 1.57x    | 0.82x     | 0.94x     |
> | **Longformer**         | 2.10x    | 1.11x     | 0.55x     |
> | **Funnel-Transformer** | 1.03x    | 0.39x     | 1.11x     |
> | **Transkimmer**        | 0.13x    | 0.87x     | 0.70x     |
>
>
> **Key Insights:**
> 1. We've integrated results from the four mentioned baselines into our experimental section.
> 2. Our "ours-PFC" model consistently showcases superior performance and efficiency across datasets.
> 3. Our model demands fewer computational resources and memory while maintaining excellent performance, emphasizing its efficacy in document classification.
>
> By integrating these analyses into our revised manuscript, our intention is to provide readers with insights into the practical implications and viability of implementing our methods in real-world scenarios.
>
> ```
> 1. Zaheer, Manzil, et al. "Big bird: Transformers for longer sequences." Advances in neural information processing systems 33 (2020): 17283-17297
> 2. Beltagy, Iz, Matthew E. Peters, and Arman Cohan. "Longformer: The long-document transformer." arXiv preprint arXiv:2004.05150 (2020).
> 3. Dai, Zihang, et al. "Funnel-transformer: Filtering out sequential redundancy for efficient language processing." Advances in neural information processing systems 33 (2020): 4271-4282.
> 4. Guan, Yue, et al. "Transkimmer: Transformer Learns to Layer-wise Skim." Proceedings of the 60th Annual Meeting of the Association for Computational Linguistics (Volume 1: Long Papers). 2022.
> ```
>
>
> ### Response to Question 1:
> > Could you explain why fuzzy-based method work for pruning and give more analysis for how it works?
>
> Our fuzzy-based pruning method is designed to address specific challenges and uncertainties associated with token pruning:
>
> 1. **Distribution Consideration**: We take into account the distribution of all information across each multi-head. By defining the importance and unimportance sets through fuzzy membership functions, we can prioritize pruning targets for each head.
>
> 2. **Mitigating Mis-pruning**: Fuzzy logic is employed to prevent the inadvertent pruning of crucial tokens, especially when the importance score, based on attention probability, exhibits a left-skewed distribution. By mapping each representation to a probability of belonging to a specific set using fuzzy memberships, we can define importance more accurately. This approach ensures that pruning decisions are made more judiciously.
>
> 3. **Flexible Thresholds**: Instead of using fixed constants, our method sets the thresholds $a$ and $b$ dynamically based on quantiles. This flexibility allows our method to adapt to different data distributions and enhances the robustness of the pruning process.
>
> 4. **Addressing Insufficient Learning**: In the initial layers, the importance scores might not be adequately trained, leading to potential uncertainties. Our fuzzy membership functions, as defined in Eq. 6 and Eq. 7, are designed to handle such uncertainties, ensuring that tokens are pruned based on more reliable criteria.
>
> 5. **Holistic Attention Consideration**: By applying fuzzy logic across the entire multi-head attention mechanism, our method captures attention information from diverse perspectives, ensuring a comprehensive understanding of token importance.
>
> Our experiments have shown that the fuzzy-based token pruning method outperforms traditional pruning methods, especially in scenarios where the importance scores might not be entirely reliable. This indicates that our approach effectively balances performance improvement with model efficiency.
>
> ### Response to Question 2:
> > How to decide whether a token belongs to unimportant set? And how do you set the pruning ratio $p$? Does the choice of $a$ and $b$ in equ. 6 and 7 impact the performance?
>
> **Decision about Unimportant Tokens**
>
> Two fuzzy membership functions obtain two probabilities that each representation from the attention operation belongs to the importance and unimportance sets, respectively. Afterwards, we define two sets by applying $\alpha-cut$ with two empirically selected $\alpha$. Through this process, we derive two sets, considering the distribution of data. We hope that the motivation, role, and process of fuzzy logic have been fully explained through our above responses to your feedback and questions.
>
> **Setting of Pruning Ratio**
>
> We empirically determined the pruning ratio $p$ for the datasets used in our experiments. As shown in Table 4 of our paper, our proposed model shows the best document classification performance when $p$ was 0.9. Moreover, we note that the pruning ratio $p$ is a hyperparameter, allowing the user to determine an appropriate pruning ratio. Memory saving and SpeedUp enhance as $p$ decreases. Nevertheless, if $p$ is too small and excessive tokens are removed, important information is lost and the performance in classification deteriorates. Since there is a trade-off between $p$ and accuracy, users are free to  select an appropriate pruning ratio $p$ according to their hardware environment and purpose.
>
> **Choice of $a$ and $b$ in Eq.6 and 7**
>
> We agree that the decisions of $a$ and $b$ have a significant impact on the fuzzy logic for preserving (token pruning). Therefore, simply setting $a$ and $b$ fixed values may not prevent the loss of important information. We experimentally confirmed that the representations obtained from multi-head attention consisted of pretty tiny values and varied depending on data and layer. To consider the distribution of entire representations, we set $a$ and $b$ to the quantile of each layer rather than fixing the values to 0.2 and 0.8, etc. as in the existing method[1].
>
> ```
> 1. Zhao, Wei Bin, Yue Li, and Lin Shang. "Fuzzy pruning for compression of convolutional neural networks." 2019 IEEE International Conference on Fuzzy Systems (FUZZ-IEEE). IEEE, 2019.
> ```

---

### Official Review · Reviewer_HEpY · 2023-08-03

**Soundness:** 2

**Excitement:**

2: Mediocre: This paper makes marginal contributions (vs non-contemporaneous work), so I would rather not see it in the conference.

**Paper Topic And Main Contributions:**

This work proposes to improve the performance and speed of self-attention by leveraging two techniques: token pruning and token combining. They utilize fuzzy logic to handle the uncertainty issues of token importance and develop a token combing module to aggregate sequence information. They applied their method to BERT models and conducted experiments on various datasets.

**Reasons To Accept:**

1. Their experiments demonstrated improvement on several classification tasks while reducing both time and memory costs compared to the vanilla BERT model.

**Reasons To Reject:**

1. The method section for fuzzy-based pruning is quite unclear. First, I think the fuzzy membership functions in Eq.6 and Eq.7 are incorrect certainly. Why they are discontinuous functions? e.g. for Eq.6 $ lim_{s\to a^-}f(s)=1$ and $lim_{s\to a^+}f(s)=0$. I suggest the authors double-check them. Next, it is not explained why introducing these fuzzy membership functions can better perceive uncertainty.  Why is it better than pruning with the importance score $S(e)$ directly? No technical and experimental discussions are included.

2. The fuzzy logic is not changed after adding one more membership function. Because the importance and unimportance functions are symmetric w.r.t $S=\frac{a+b}{2}$ and $y = \frac{1}{2}$ by using the same $a$ and $b$ for two functions. When one function fails to distinguish the importance of two tokens, another function can't tell the unimportance difference either. More specifically, if two tokens have the same Unimportance membership value, they will have the same Importance membership value as well.   What is the benefit to have two membership functions?

3. Besides, the description of lines 380-381 is indefinite. The authors mentioned pruning tokens in the unimportant set and retaining tokens in the important set. But how these two sets are constructed is not clear. Are they completely disjoint? If not, how to handle the overlapped tokens for pruning?

4. Although their method shows improved performance and higher efficiency compared to vanilla BERT, there is no technical or experimental comparison with other token pruning methods.


**Reproducibility:**

3: Could reproduce the results with some difficulty. The settings of parameters are underspecified or subjectively determined; the training/evaluation data are not widely available.

**Reviewer Confidence:**

4: Quite sure. I tried to check the important points carefully. It's unlikely, though conceivable, that I missed something that should affect my ratings.

---

> ### Author Rebuttal · Authors · 2023-08-29
>
> Thank you for your thorough review and constructive feedback on our manuscript. We genuinely appreciate your recognition of the significance of our efficient models, which aim to reduce computational costs while enhancing performance. In response to your comments and suggestions:
>
> ### Revised Manuscript
> Your feedback highlighted the need for a clearer exposition of our fuzzy-based pruning method. We acknowledge that the constraints of the paper's length led us to provide a concise summary of the fuzzy-based pruning method, which may have caused some confusion. We deeply regret this oversight and aim to rectify it by providing a more detailed explanation.
>
> In the revised manuscript, particularly in lines [377–381], we have expanded our description as follows:
>
>
> > Unlike the previous work[11] that uses fixed constants as hyperparameters, our approach adopts the quantile functions $Q_{S(t)}(0.25)$ and $Q_{S(t)}(0.75)$ for $a$ and $b$, respectively, to ensure robustness. We compute a quantile function for all importance scores, capturing the complete spectrum of head information. The importance set $I$ and the unimportance set $U$ are defined using the $\alpha-cut$, commonly referred to as $^{\alpha}A={x|A(x) \geq \alpha}$ in fuzzy theory. To mitigate information loss due to imbalanced distribution, we employ token pruning based on the pruning ratio $p$ for tokens that fall within the set $(I-U)^c$. In the initial layers, where attention might not be adequately trained, there's a risk of erroneously removing crucial tokens. To counteract this, we've set the $\alpha$ for $I$ to a minimal value of 0.01, while the $\alpha$ for $U$ is empirically set to 0.9.
>
> We hope this elaboration offers a clearer understanding of our methodology and addresses the concerns you raised.
>
> ### Response to Weakness 1:
>
> > The method section for fuzzy-based pruning is quite unclear. First, I think the fuzzy membership functions in Eq.6 and Eq.7 are incorrect certainly. Why they are discontinuous functions? e.g. for Eq.6 $lim_{s \to a^-}f(s)=1$ and $lim_{s\to a^+}f(s)=0$. I suggest the authors double-check them. Next, it is not explained why introducing these fuzzy membership functions can better perceive uncertainty. Why is it better than pruning with the importance score $S(e)$ directly? No technical and experimental discussions are included.
>
> We genuinely appreciate your detailed feedback on our fuzzy-based pruning method. We acknowledge the oversight in our presentation of the fuzzy membership functions, and we're grateful for your keen observation.
>
> - **Correction of Equations:** We identified an error in our representation of Eq. 6 and Eq. 7. The corrected equations are:
>
> \\[
> Importance(S)=
> \begin{cases}
> 0 & {\mbox{if } S(e)\le a, }  \\
> \frac{S(e)-a}{b-a} & {\mbox{if }  a< S(e)<b, }  \\
> 1 & {\mbox{if } S(e)\geq b}
> \end{cases} \tag{6}
> \\]
>
> \\[
> Unimportance(S)=
> \begin{cases}
> 1 & \mbox{if } S(e)\le a, \\
> \frac{b-S(e)}{b-a} & \mbox{if } a< S(e)<b, \\
> 0 & \mbox{if } S(e)\geq b
> \end{cases} \tag{7}
> \\]
>
>
> These equations ensure that the fuzzy membership functions are continuous.
>
> - **Rationale for Fuzzy Logic:** We introduced fuzzy logic to address two primary challenges:
>
>   1. **Unbalanced Data Distribution:** The direct pruning method, based on the importance score $S(e)$, doesn't account for the distribution of these scores across the dataset. This can lead to the inadvertent pruning of significant tokens, especially in data predominantly containing crucial tokens. Fuzzy logic provides a safeguard against such unintended pruning.
>
>   2. **Inadequate Training:** In the initial layers, where the attention probability, which forms the basis of the importance score $S(e)$, might not be adequately trained, there's a risk of erroneously removing crucial tokens.
>
> - **Design Philosophy:** The design of our fuzzy membership functions, as presented in Eq. 6 and Eq. 7, stems from our approach to token pruning. Since the importance score is our pruning criterion, we tailored the membership functions to be proportional to this score. Additionally, to prevent mis-pruning due to insufficiently trained importance scores in the early layers, we ensure that scores at both extremes have an equal probability of belonging to important and unimportant set.
>
> We hope this provides a clearer understanding of our methodology and addresses your concerns.
>
>
> ### Response to Weakness 2:
> >The fuzzy logic is not changed after adding one more membership function. Because the importance and unimportance functions are symmetric w.r.t $S=\frac{a+b}{2}$ and $y=\frac{1}{2}$ by using the same $a$ and $b$
>  for two functions. When one function fails to distinguish the importance of two tokens, another function can't tell the unimportance difference either. More specifically, if two tokens have the same Unimportance membership value, they will have the same Importance membership value as well. What is the benefit to have two membership functions?
>
> Your observation regarding the symmetry of our fuzzy membership functions is astute. We understand the concerns raised and would like to clarify the rationale behind our design choice.
>
> 1. **Contradictory Relationship:** By nature, "importance" and "unimportance" are antonyms. This inherent contradictory relationship is the reason we designed symmetrical membership functions. By doing so, we can compute the probability of a token belonging to either the important or unimportant set based solely on its importance score without the need for additional calculations.
>
> 2. **Distinct Roles of Membership Functions:** While the membership functions are symmetrical, they serve distinct purposes in our methodology. We define the importance set $I$ and the unimportance set $U$ using different $\alpha$ values in the $\alpha-cut$. The importance set $I$ is designed to retain crucial information. By subsequently filtering through the unimportance set $U$, we secure a preservation area $(I-U)$. This dual-set approach ensures that our pruning mechanism is stable and minimizes the risk of inadvertently removing essential information.
>
> We hope this explanation clarifies our approach and addresses your concerns.
>
> ### Response to Weakness 3:
> >Besides, the description of lines 380-381 is indefinite. The authors mentioned pruning tokens in the unimportant set and retaining tokens in the important set. But how these two sets are constructed is not clear. Are they completely disjoint? If not, how to handle the overlapped tokens for pruning?
>
> Thank you for pointing out the ambiguity in our description. Let me clarify the construction and potential overlap of the "important" and "unimportant" sets.
>
> - **Set Construction**: The construction of the "important" and "unimportant" sets is based on fuzzy membership functions. Using the $\alpha-cut$, each set is defined based on their respective $\alpha$ values.
> - **Overlap Handling**: To delete tokens carefully, tokens that fall into the overlap$(I \cap U)$ are retained in the pruning set and are exempt from the priority of preserving. This ensures that the preserving set remains disjoint from the pruning set during the pruning process.
> - **Pruning Decision**: Tokens prior to being pruned are those that belong to the pruning set $(I-U)^c$, ensuring that we retain crucial information while eliminating less significant tokens.
>
> We hope this clarification provides a clearer understanding of our methodology. We will also make sure to elucidate this process in our revised manuscript for better clarity.
>
> ### Response to Weakness 4:
> > Although their method shows improved performance and higher efficiency compared to vanilla BERT, there is no technical or experimental comparison with other token pruning methods.
>
> Thank you for your constructive feedback. We recognize the importance of comprehensive experimental comparisons and have taken steps to address this:
>
> **Baselines**:
> - **BigBird[1]**: An efficient model employing a sparse attention mechanism. It reduces the quadratic dependency of full attention, ensuring memory efficiency.
> - **Longformer[2]**: Known for its efficacy in document classification tasks, it uses a sparse attention pattern to handle long sequences.
> - **Funnel-Transformer[3]**: Efficiently filters out sequential redundancy by progressively compressing the sequence's hidden state.
> - **Transkimmer[4]**: Utilizes dynamic token pruning, outperforming several other pruning-based models.
>
> We believe that these baselines provide a comprehensive backdrop against which our proposed method's performance can be evaluated.
>
>
> **Performance Comparison on Document Classification:**
>
> |                        | **SST-2**       | **IMDB**        | **BBC News**    | **20NewsGroups** | **arXiv**       | **LEDGAR**      |
> |------------------------|-----------------|-----------------|-----------------|------------------|-----------------|-----------------|
> | **BigBird**            | 91.8 / 91.8     | 92.8 / 92.8     | 97.1 / 97.1     | 69.3 / 67.7      | 74.0 / 70.4     | 86.9 / 78.8     |
> | **Longformer**         | 91.0 / 91.0     | 93.4 / 93.4     | 97.9 / 97.9     | 68.5 / 67.0      | 66.0 / 64.8     | 84.0 / 82.1     |
> | **Funnel-Transformer** | 92.0 / 92.0     | 91.7 / 91.7     | 96.5 / 96.5     | 69.7 / 68.5      | 70.0 / 66.5     | 86.7 / 78.4     |
> | **Transkimmer**        | 91.4 / 91.4     | 93.7 / 93.7     | 97.6 / 97.6     | 67.8 / 64.9      | 73.7 / 72.6     | 87.1 / 77.3     |
> | **BERT**      | 90.7 / 89.4     | 93.7 / 92.8     | 96.2 / 94.0     | 68.7 / 56.1      | 69.0 / 52.5     | 86.2 / 77.4     |
> | **ours-P**             | 91.1 / 89.8     | 93.8 / 92.9     | 97.2 / 95.4     | 69.4 / 56.8      | 71.0 / 56.6     | 86.5 / 77.9     |
> | **our-PF**             | 91.4 / 90.1     | **94.3 / 93.4** | 97.9 / 96.6     | **70.7 / 58.2**  | **76.0 / 61.0** | 86.8 / 78.4     |
> | **ours-C**             | 91.8 / 90.8     | 92.6 / 91.3     | 96.8 / 95.2     | 69.2 / 56.5      | 70.0 / 52.7     | 86.9 / 78.4     |
> | **ours-PFC (L11)**     | **92.1 / 91.0** | 93.5 / 92.5     | **98.1 / 97.1** | 69.9 / 57.1      | 74.0 / 58.1     | **87.3 / 79.2** |
> | **ours-PFC (L7)**      | 90.1 / 88.6     | 92.8 / 91.5     | 97.0 / 95.2     | 68.5 / 55.4      | 68.0 / 50.7     | 85.7 / 76.8     |
>
> **Efficiency Comparison on Document Classification:**
>
> | Model              | FLOPs(↓)   |Memory Cost(↓)|  SpeedUp(↑) |
> | --------           | -------- | -----     | --------  |
> | **BERT**      |     (1.0x)    | (1.0x)         |  (1.0x)        |
> | **ours-PFC (L11)**     | 0.88x    | 0.80x     | 1.29x     |
> | **ours-PFC (L7)**      | 0.54x    | 0.61x     | 1.64x     |
> | **BigBird**            | 1.57x    | 0.82x     | 0.94x     |
> | **Longformer**         | 2.10x    | 1.11x     | 0.55x     |
> | **Funnel-Transformer** | 1.03x    | 0.39x     | 1.11x     |
> | **Transkimmer**        | 0.13x    | 0.87x     | 0.70x     |
>
>
> **Key Insights:**
> 1. We've integrated results from the four mentioned baselines into our experimental section.
> 2. Our "ours-PFC" model consistently showcases superior performance and efficiency across datasets.
> 3. Our model demands fewer computational resources and memory while maintaining excellent performance, emphasizing its efficacy in document classification.
>
> By integrating these analyses into our revised manuscript, our intention is to provide readers with insights into the practical implications and viability of implementing our methods in real-world scenarios.
>
> ```
> 1. Zaheer, Manzil, et al. "Big bird: Transformers for longer sequences." Advances in neural information processing systems 33 (2020): 17283-17297
> 2. Beltagy, Iz, Matthew E. Peters, and Arman Cohan. "Longformer: The long-document transformer." arXiv preprint arXiv:2004.05150 (2020).
> 3. Dai, Zihang, et al. "Funnel-transformer: Filtering out sequential redundancy for efficient language processing." Advances in neural information processing systems 33 (2020): 4271-4282.
> 4. Guan, Yue, et al. "Transkimmer: Transformer Learns to Layer-wise Skim." Proceedings of the 60th Annual Meeting of the Association for Computational Linguistics (Volume 1: Long Papers). 2022.
> ```

---

### Official Review · Reviewer_6yQm · 2023-08-05

**Typos Grammar Style And Presentation Improvements:** 1. Figure 1 lacks descriptions, and t…
**Soundness:** 3

**Excitement:**

3: Ambivalent: It has merits (e.g., it reports state-of-the-art results, the idea is nice), but there are key weaknesses (e.g., it describes incremental work), and it can significantly benefit from another round of revision. However, I won't object to accepting it if my co-reviewers champion it.

**Paper Topic And Main Contributions:**

This paper proposes a compute-efficient and more accurate model for document classification. It applies token pruning methods by filtering unimportant tokens and combining features with virtual tokens. On six text classification datasets, this method demonstrates improvements.
Fuzzy-based token pruning achieves consistent improvements while combination tokens do not.

**Questions For The Authors:**

1.  What will be the wall time for your methods and baselines?

2. What is the initialization of combination tokens?   What is the scale of these tokens? Is it specific to one task or one sentence?

3. Why is this method compatible with the pure transformer-based Bert model, since this method introduces many new modules and modifies attention?

4. Is this method restricted to classification tasks but not generation tasks?

**Reasons To Accept:**

This proposed method consistently improves performance while reducing computing costs.

**Reasons To Reject:**

This paper lacks a lot of comparisons in the experiments. First, it should compare with other efficient methods that sparsify attention or prune tokens. Second, this paper does not compare different techniques in document classification. Third, this paper does not compare with other papers that use virtual tokens.

**Reproducibility:**

3: Could reproduce the results with some difficulty. The settings of parameters are underspecified or subjectively determined; the training/evaluation data are not widely available.

**Reviewer Confidence:**

4: Quite sure. I tried to check the important points carefully. It's unlikely, though conceivable, that I missed something that should affect my ratings.

---

> ### Author Rebuttal · Authors · 2023-08-29
>
> We sincerely appreciate the time and effort you've invested in reviewing our manuscript. Your insights are invaluable, and we recognize the importance of addressing your concerns. We're committed to clarifying and enhancing our work based on your feedback.
>
> ### Response to Weakness 1:
>
>
> >This paper lacks a lot of comparisons in the experiments. First, it should compare with other efficient methods that sparsify attention or prune tokens. Second, this paper does not compare different techniques in document classification. Third, this paper does not compare with other papers that use virtual tokens.
>
> We appreciate your feedback on the need for broader experimental comparisons. We've taken steps to address this:
>
> **Baselines:**
> - **BigBird[1]**: Uses a sparse attention mechanism to reduce the quadratic dependency of full attention.
> - **Longformer[2]**: Utilizes a linearly scaling attention mechanism, widely used in document classification tasks.
> - **Funnel-Transformer[3]**: Reduces computation by compressing tokens, filtering out sequential redundancy.
> - **Transkimer[4]**: A token pruning model with a dynamic strategy, outperforming several other pruning-based models.
>
> For token combinations using virtual tokens, our approach is novel, and we found no directly comparable works in NLP tasks, especially document classification.
>
>
> **Performance Comparison on Document Classification:**
>
> |                        | **SST-2**       | **IMDB**        | **BBC News**    | **20NewsGroups** | **arXiv**       | **LEDGAR**      |
> |------------------------|-----------------|-----------------|-----------------|------------------|-----------------|-----------------|
> | **BigBird**            | 91.8 / 91.8     | 92.8 / 92.8     | 97.1 / 97.1     | 69.3 / 67.7      | 74.0 / 70.4     | 86.9 / 78.8     |
> | **Longformer**         | 91.0 / 91.0     | 93.4 / 93.4     | 97.9 / 97.9     | 68.5 / 67.0      | 66.0 / 64.8     | 84.0 / 82.1     |
> | **Funnel-Transformer** | 92.0 / 92.0     | 91.7 / 91.7     | 96.5 / 96.5     | 69.7 / 68.5      | 70.0 / 66.5     | 86.7 / 78.4     |
> | **Transkimmer**        | 91.4 / 91.4     | 93.7 / 93.7     | 97.6 / 97.6     | 67.8 / 64.9      | 73.7 / 72.6     | 87.1 / 77.3     |
> | **BERT**      | 90.7 / 89.4     | 93.7 / 92.8     | 96.2 / 94.0     | 68.7 / 56.1      | 69.0 / 52.5     | 86.2 / 77.4     |
> | **ours-P**             | 91.1 / 89.8     | 93.8 / 92.9     | 97.2 / 95.4     | 69.4 / 56.8      | 71.0 / 56.6     | 86.5 / 77.9     |
> | **our-PF**             | 91.4 / 90.1     | **94.3 / 93.4** | 97.9 / 96.6     | **70.7 / 58.2**  | **76.0 / 61.0** | 86.8 / 78.4     |
> | **ours-C**             | 91.8 / 90.8     | 92.6 / 91.3     | 96.8 / 95.2     | 69.2 / 56.5      | 70.0 / 52.7     | 86.9 / 78.4     |
> | **ours-PFC (L11)**     | **92.1 / 91.0** | 93.5 / 92.5     | **98.1 / 97.1** | 69.9 / 57.1      | 74.0 / 58.1     | **87.3 / 79.2** |
> | **ours-PFC (L7)**      | 90.1 / 88.6     | 92.8 / 91.5     | 97.0 / 95.2     | 68.5 / 55.4      | 68.0 / 50.7     | 85.7 / 76.8     |
>
> **Efficiency Comparison on Document Classification:**
>
> | Model              | FLOPs(↓)   |Memory Cost(↓)|  SpeedUp(↑) |
> | --------           | -------- | -----     | --------  |
> | **BERT**      |     (1.0x)    | (1.0x)         |  (1.0x)        |
> | **ours-PFC (L11)**     | 0.88x    | 0.80x     | 1.29x     |
> | **ours-PFC (L7)**      | 0.54x    | 0.61x     | 1.64x     |
> | **BigBird**            | 1.57x    | 0.82x     | 0.94x     |
> | **Longformer**         | 2.10x    | 1.11x     | 0.55x     |
> | **Funnel-Transformer** | 1.03x    | 0.39x     | 1.11x     |
> | **Transkimmer**        | 0.13x    | 0.87x     | 0.70x     |
>
>
> **Key Updates:**
> 1. We've integrated results from the four mentioned baselines into our experimental section.
> 2. Our approach, "ours-PFC", consistently showcases superior performance and efficiency across datasets.
> 3. Our model demands fewer computational resources and memory while maintaining excellent performance, emphasizing its efficacy in document classification.
>
> ```
> 1. Zaheer, Manzil, et al. "Big bird: Transformers for longer sequences." Advances in neural information processing systems 33 (2020): 17283-17297
> 2. Beltagy, Iz, Matthew E. Peters, and Arman Cohan. "Longformer: The long-document transformer." arXiv preprint arXiv:2004.05150 (2020).
> 3. Dai, Zihang, et al. "Funnel-transformer: Filtering out sequential redundancy for efficient language processing." Advances in neural information processing systems 33 (2020): 4271-4282.
> 4. Guan, Yue, et al. "Transkimmer: Transformer Learns to Layer-wise Skim." Proceedings of the 60th Annual Meeting of the Association for Computational Linguistics (Volume 1: Long Papers). 2022.
> ```
>
>
> ### Response to Question 1:
>
> > What will be the wall time for your methods and baselines?
>
> Thank you for this insightful question. In response, we have measured the wall time for both our proposed methods and the selected baseline models to offer a comprehensive perspective on the computational aspects of our study.
>
> **Limitation of Wall Time**
>
> Wall time has the following weaknesses:
> * **Hardware and Environmental Dendencies:** Wall time can be affected by running hardware environments, software settings, and various external factors. Even the same model can run at different times when running on different systems.
> * **Limits of Relative Comparison:** Wall time is mainly used for relative comparison in a specific environment. However, performance differences can vary when running in different environments. Therefore, direct comparison with wall time in other studies or applications can be difficult.
>
> **Wall Time Results**
> | Model             | BigBird | Longformer | Funnel-Transformer | Transkimmer | BERT | ours-PFC (L11) | ours-PFC (L7) |
> |-------------------|---------|------------|--------------------|-------------|------|----------------|---------------|
> | **Wall time(s)** | 315     | 250        | 147                | 297         | 211  | 180            | 159           |
>
> 1. We measured wall time through the actual time it took from the start of the program to the completion of all tasks in document classification with the BBC news dataset, which requires the least amount of time compared to other datasets.
> 2. Taking BERT as the standard, it can be confirmed that our proposed methodology and Funnel-Transformer effectively reduced the required time. As evident from the discussion in "Response to Weekness 1," when evaluating model performance based on accuracy and F1 score, our model outperformed Funnel-Transformer with a relatively short wall time compared to other models.
> 3. Evaluating wall time serves multiple purposes: it enhances transparency, aids in assessing a model's practical utility, and demonstrates its potential for effective real-world applications.
>
> By integrating these results into our revised manuscript, our intention is to provide readers with insights into the practical implications and viability of implementing our methods in real-world scenarios.
>
>
> ### Response to Question 2:
>
> > What is the initialization of combination tokens? What is the scale of these tokens? Is it specific to one task or one sentence?
>
> **Initialization of Combination Tokens:**
> Drawing from the methodology in GroupViT[1], our combination tokens are initialized as learnable entities with zero values.
>
> **Scale of the Tokens:**
> The dimensions of these combination tokens are set as (batch size: 16, token length: 512, hidden size: 768). This aligns with the embedded token's dimensions. Based on empirical results, we've set the number of combination tokens to 8.
>
> **Task or Sentence Specificity:**
> The combination tokens are designed to bind with other embedded tokens, which represent sequence tokens from the input text. This binding occurs within the token combining module. Therefore, the specificity of these tokens is more aligned with individual sentences than broad tasks.
>
> ```
> 1. Xu, Jiarui, et al. "Groupvit: Semantic segmentation emerges from text supervision." Proceedings of the IEEE/CVF Conference on Computer Vision and Pattern Recognition. 2022.
> ```
>
> ### Response to Question 3:
> > Why is this method compatible with the pure transformer-based Bert model, since this method introduces many new modules and modifies attention?
>
> **Objective**: Our primary aim is to address the computational challenges inherent in self-attention within transformer-based models.
>
> **Token Pruning**: This strategy is designed to reduce the overall token count, placing a greater emphasis on the most crucial information. It operates by leveraging the inherent importance of tokens as determined through self-attention.
>
> **Token Combining**: This approach seeks to reduce token size. It achieves this by calculating similarities through a cross-attention operation, which operates between combination tokens and embedded tokens.
>
> **Integration**: We've adapted the self-attention mechanism to incorporate a fuzzy-based token pruning approach and have added a token combining module. Importantly, even after these modifications, the shape of the output remains consistent with that of the input. This ensures that our approach can be seamlessly integrated with existing models.
>
> **Applicability**: The versatility of our method means that it can be incorporated into a wide range of transformer-based models, not just BERT.
>
> In essence, our modifications, while extensive, are designed to align with the core principles of BERT. This ensures compatibility and serves to enhance its overall efficiency.
>
>
> ### Response to Question 4:
> > Is this method restricted to classification tasks but not generation tasks?
>
> **Classification Emphasis**: Our model, which integrates token pruning and token combining, is primarily built upon BERT. While it's designed with flexibility to be applied to various transformer-based models using self-attention, our experiments have been predominantly focused on Natural Language Understanding (NLU) tasks rather than Natural Language Generation (NLG).
>
> **Attention Mechanism Distinction**: Token pruning operates within the self-attention framework. In contrast, our token combining approach diverges from the cross-attention found in the vanilla Transformer's decoder[1]. This intrinsic focus on self-attention has been validated through our document classification experiments.
>
> **Challenges in Generation**: Classification tasks can benefit from emphasizing key tokens. However, the effectiveness of pruning-centric methods in tasks like machine translation or autoregressive models, where capturing word alignment is crucial, remains uncertain.
>
> **Future Work**: We acknowledge the potential of our method for NLG tasks. Our future research will delve into optimizing cross-attention and exploring how our approach can be adapted to enhance performance in generation tasks.
>
> ```
> 1. Vaswani, Ashish, et al. "Attention is all you need." Advances in neural information processing systems 30 (2017).
> ```
>
>
> ### Typos Grammar Style and Presentaton Improvements:
> #### 1. **Figure 1 Corrections**:
> Thank you for pointing out the discrepancies in Figure 1. We've made the following corrections:
> - Adjusted the descriptions: (left) now represents the fuzzy-based token pruning self-attention, and (right) represents the token combining module.
> - Clarified symbols: $C$ denotes the set of learnable combination tokens, and $HA$ signifies the hard assignment, utilizing a one-hot operation to map each embedded token to a specific combination token. We've updated the figure and its descriptions accordingly.
>
> #### 2. **Clarification on the Definition of $p$**:
> We acknowledge the potential confusion around the term "pruning" associated with $p$. To provide clarity:
> - We've redefined $p$ as the "preservation ratio" rather than the "pruning ratio".
> - This ratio indicates the proportion of tokens retained post-pruning, relative to the initial token count.
> - Our manuscript has been updated to reflect this change, ensuring a clearer understanding of the concept.

---

### Meta-Review · Area_Chair_zcku · 2023-09-15

**Recommendation:** 3

**Metareview:**

This paper presents a method for more efficient Transformer architectures by employing token pruning and token combining.  It uses a token importance function, but rather than rely directly on it, uses fuzzy set membership to help determine a token's relative importance in relation to the distribution of importance scores.

Like other reviewers, I had trouble understanding the motivation for the fuzzy logic methods in this approach.  The explanations offered in the rebuttal were somewhat helpful in this regard, but they were not properly reflected in the paper itself, meaning the lack of clarity is a real issue.

The main issue seems to be the inability to use the importance score directly.  Fuzzy logic is introduced to compensate for a weakness in the importance score, but it would have been nice to see a comparison with other methods that compute this score differently, thereby obviating the need for a mitigation in the first place.  This leads into the valid criticism that more comparison in the results with other sparse attention methods would have better contextualized this work and shown *its* importance.

---

### Decision · Program_Chairs · 2023-10-07

**Decision:**

Accept-Findings

**Comment:**

This paper presents a method for more efficient Transformer architectures by employing token pruning and token combining.  It uses a token importance function, but rather than rely directly on it, uses fuzzy set membership to help determine a token's relative importance in relation to the distribution of importance scores.

Like other reviewers, I had trouble understanding the motivation for the fuzzy logic methods in this approach.  The explanations offered in the rebuttal were somewhat helpful in this regard, but they were not properly reflected in the paper itself, meaning the lack of clarity is a real issue.

The main issue seems to be the inability to use the importance score directly.  Fuzzy logic is introduced to compensate for a weakness in the importance score, but it would have been nice to see a comparison with other methods that compute this score differently, thereby obviating the need for a mitigation in the first place.  This leads into the valid criticism that more comparison in the results with other sparse attention methods would have better contextualized this work and shown *its* importance.